# Mutation rate dynamics reflect ecological change in an emerging zoonotic pathogen

**Gemma G. R. Murray**[1]*, **Andrew J. Balmer**[1], **Josephine Herbert**[1¤a], **Nazreen F. Hadjirin**[1], **Caroline L. Kemp**[1], **Marta Matuszewska**[1], **Sebastian Bruchmann**[1], **A. S. Md. Mukarram Hossain**[1¤b], **Marcelo Gottschalk**[2], **Alexander W. Tucker**[1], **Eric Miller**[1☯¤c], **Lucy A. Weinert**[1☯]

**1** Department of Veterinary Medicine, University of Cambridge, Cambridge, United Kingdom, **2** Département de Pathologie et Microbiologie, Université de Montréal, Montréal, Canada

☯ These authors contributed equally to this work.
¤a Current address: School of Pharmacy and Biomedical Science, University of Portsmouth, Portsmouth, United Kingdom
¤b Current address: Cancer Biomarker Centre, Cancer Research UK Manchester Institute, The University of Manchester, Alderley Park, United Kingdom
¤c Current address: Haverford College, Pennsylvania, United States of America
* ggrmurray@gmail.com

**Data Availability Statement:** All sequence data from the two experiments have been deposited on the short read archive under the BioProject ID: PRJNA763153. Assemblies of the eight ancestral

## Abstract

Mutation rates vary both within and between bacterial species, and understanding what drives this variation is essential for understanding the evolutionary dynamics of bacterial populations. In this study, we investigate two factors that are predicted to influence the mutation rate: ecology and genome size. We conducted mutation accumulation experiments on eight strains of the emerging zoonotic pathogen *Streptococcus suis*. Natural variation within this species allows us to compare tonsil carriage and invasive disease isolates, from both more and less pathogenic populations, with a wide range of genome sizes. We find that invasive disease isolates have repeatedly evolved mutation rates that are higher than those of closely related carriage isolates, regardless of variation in genome size. Independent of this variation in overall rate, we also observe a stronger bias towards G/C to A/T mutations in isolates from more pathogenic populations, whose genomes tend to be smaller and more AT-rich. Our results suggest that ecology is a stronger correlate of mutation rate than genome size over these timescales, and that transitions to invasive disease are consistently accompanied by rapid increases in mutation rate. These results shed light on the impact that ecology can have on the adaptive potential of bacterial pathogens.

## Author summary

Mutations are the ultimate source of all genetic variation and mutation rates vary considerably both within and between bacterial species. Understanding the drivers of this variation is important as it influences the capacity of bacteria to respond to challenges. It is particularly important for bacterial pathogens as it impacts how they respond to host immune responses and antibiotic treatments. Our study investigates how mutation rates

strains have been deposited under the BioProject ID: PRJNA763404. Accessions are provided in S1 Table.

**Funding:** This work was primarily funded by an Isaac Newton Trust / Wellcome Trust ISSF / University of Cambridge Joint Research Grant to LAW. In addition, LAW, ELM, GGRM were supported by a Sir Henry Dale Fellowship to LAW jointly funded by the Wellcome Trust and the Royal Society (109385/Z/15/Z). GGRM was also supported by a Research Fellowship at Newnham College, University of Cambridge, and by a ZELS BBSRC award (BB/L018934/1) to AWT. AJB was supported by a BBSRC-funded studentship. MM was supported by a Medical Research Council studentship, co-funded by the Raymond and Beverly Sackler Fund. MH was supported by EU Horizon 2020 grant "PIGSs" (no. 727966) to LAW and AWT, which also funded the collection of some of the isolates in this study. The funders had no role in study design, data collection and analysis, decision to publish, or preparation of the manuscript.

**Competing interests:** The authors have declared that no competing interests exist.

vary within a bacterial species that has both a variable genome size and a variable ecological relationship with its host. While inter-species comparisons have found that bacterial species with smaller genomes tend to have faster mutation rates, our within-species comparisons show no evidence of a link between mutation rate and genome size. Instead, we find that strains that were involved in invasive infections have faster mutation rates than those carried asymptomatically by a host. This suggests that different factors influence mutation rate variation over different timescales, and that short-term changes are sensitive to ecological transitions. This contributes to our understanding of both the adaptive potential of pathogens, and the obstacles that bacteria have to overcome to cause disease in their host.

## Introduction

Mutation rates vary within and between bacterial species, contributing to differences in both the burden of deleterious mutations and the capacity to adapt to environmental change [1–3]. Understanding how mutation rates evolve in response to selective pressures is fundamental to our understanding of evolutionary dynamics. In the case of bacterial pathogens, it is important for understanding pathogen emergence, evasion of host immunity, and evolution of antimicrobial resistance [4–7].

There are several reasons to predict that bacterial pathogens will have higher mutation rates than non-pathogens. First, pathogens may be less able to constrain the evolution of their mutation rate. Mutation rates evolve as linked traits, and as deleterious mutations are more common than beneficial mutations, theory predicts that selection against these mutations will generally lead to indirect selection for lower mutation rates. Common features of pathogen ecologies such as host-restriction, frequent between-host transmission, and rapid within-host adaptation, could all contribute to a smaller effective population size [4, 8]. This could lead to a reduced efficacy of natural selection, and therefore to maladaptive evolution, including a higher mutation rate [1, 9, 10]. Second, a higher mutation rate may be associated with a lower selective cost for pathogens. Bacterial pathogens tend to have smaller genomes with fewer genes than closely related non-pathogens [11, 12], and a smaller genome will mean that a higher (per site) mutation rate will lead to fewer additional mutations [13]. Finally, pathogens might actually benefit from higher mutation rates. Pathogens often face challenging and hostile environments, and higher mutation rates can increase the speed of adaptation [3, 14, 15]. This might lead to positive selection for a higher mutation rate, although indirectly via linkage with beneficial mutations.

Despite these predictions, the influence of ecology and genome size on mutation rate dynamics in natural populations of bacterial pathogens is not well understood. Across bacterial species, mutation rates have been found to be inversely correlated with genome size, and largely independent of ecology [1, 16]. Within several bacterial species, hypermutable strains with orders of magnitude higher mutation rates than the species average have been identified [17–23]. Nevertheless, this within-species variation remains poorly characterized, and while there is evidence that transient increases in mutation rates can promote adaptation, it remains unclear whether there is an association with pathogenic ecologies.

To investigate the influence of ecology and genome size on mutation rate, we carried out mutation accumulation (MA) experiments and whole-genome sequencing for eight isolates of the opportunistic and emerging bacterial pathogen *Streptococcus suis*. This experimental approach allowed us to obtain precise estimates of mutation rate, and therefore to investigate

small-scale rate variation, which is more likely to persist over time, and to be associated with longer-term ecological transitions [24, 25]. *S. suis* both asymptomatically colonises the upper respiratory tract of pigs, and causes severe invasive infections in both pigs and humans [26, 27]. Natural variation within this species allows us to compare the mutation rates of closely related isolates with different pathogenic ecologies and a range of genome sizes (1.97–2.67 Mb) [12, 28].

## Results

We first estimated the mutation rates of two pairs of closely related isolates, that span the known range of *S. suis* genome size (strains 1–4 in Figs 1 and S1, and Tables 1 and S1). In each pair, one isolate was sampled from the site of an invasive infection (disease) and the other from the tonsils of a pig without *S. suis* associated disease (carriage). The relationship between these isolates, and their placement in a core genome phylogeny of *S. suis* was established in a previous study [12]. This study found that while both asymptomatic carriage isolates and invasive disease isolates are present across the *S. suis* phylogeny, invasive disease isolates are more common in one clade (a "more pathogenic" group) and carriage isolates more common outside of this clade (a "less pathogenic" group). Genetic diversity in the more pathogenic group is lower than, and falls within the diversity of the less pathogenic group, consistent with a more recent origin [12]. One of our pairs of isolates was sampled from this more pathogenic group (isolates 1 and 2) and the other from a less pathogenic group (isolates 3 and 4) (Figs 1A and S1). Our choice of strains therefore allows us to discriminate between three possible correlates of mutation rate: short-term transitions from carriage to disease (Fig 1B), long-term changes in pathogenicity (Fig 1C), and genome size (Fig 1D).

We estimated the mutation rates of the four isolates through four parallel 200-day MA experiments. We evolved 75 replicate lines of each strain over 200 days, with daily passaging through single-colony bottlenecks. We estimated that this 200-day period corresponds to between 3,445 to 3,991 generations for these four strains (Tables 1 and S2, and S2 Fig). We

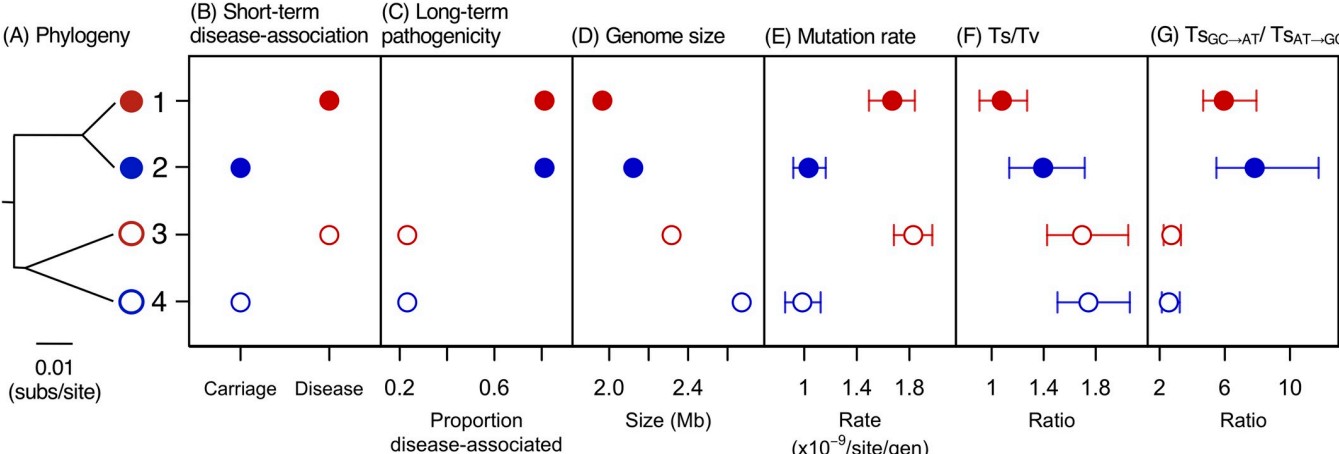

**Fig 1. Differences in mutation rate between carriage and disease strains, and in mutational spectrum between more and less pathogenic groups.** (A)-(D) The attributes of the strains in the 200-day MA experiment, and (E)-(G) our estimates of mutation rate and spectrum. (A) A phylogeny based on an alignment of shared genes. (B) Short-term disease association: whether the strain was isolated from the tonsils of animals without disease (carriage) or the site of an invasive infection (disease). (C) Long-term differences in pathogenicity between the two main groups of *S. suis* within which the two pairs of strains fall in a core genome phylogeny (based on an analysis of a global collection of *S. suis* isolates [12]). (D) Genome sizes. (E) Estimates of genome-wide single-base mutation rates. (F) Estimates of the ratio of transitions (Ts) to transversions (Tv). (G) Estimates of the ratio of G/C to A/T to A/T to G/C transitions. Disease strains (1 and 3) are shown in red and carriage strains (2 and 4) in blue. Strains from the more pathogenic group (1 and 2) are shown as filled circles, and strains from the less pathogenic group (3 and 4) are shown as empty circles. In (E)-(G) all points represent mean values across 50 replicate lines, and bars represent 95% confidence intervals estimated from bootstrapping across lines.

**Table 1. Characteristics of the eight strains of *Streptococcus suis* used in both the 200-day (not shaded) and 25-day (shaded) MA experiments and estimates of their mutation rates.**

| Group | | More pathogenic | | | | | | Less pathogenic | |
|---|---|---|---|---|---|---|---|---|---|
| Strain | 1 | 5 | 6 | 2 | 7 | 8 | 3 | 4 |
| Disease-association | Disease | Carriage | Disease | Carriage | Disease | Carriage | Disease | Carriage |
| Genome size (Mb) | 1.97 | 2.02 | 2.08 | 2.12 | 2.23 | 2.16 | 2.32 | 2.67 |
| Duration of experiment (days) | 200 | 25 | 25 | 200 | 25 | 25 | 200 | 200 |
| Number of generations | 3,991 | 523 | 513 | 3,989 | 511 | 503 | 3,484 | 3,445 |
| Single-base mutation rate (95% CI) x10⁻⁹ /site/generation | 1.67 (1.51–1.84) | 1.24 (0.58–2.05) | 1.44 (0.86–2.16) | 1.03 (0.92–1.15) | 1.43 (0.75–2.24) | 0.17 (0.01–0.42) | 1.83 1.68–1.97) | 0.99 (0.84–1.12) |

then sequenced the genomes of 50 randomly selected evolved lines of each strain, and estimated mutation rates through identifying differences in the genomes of the evolved lines compared to the genomes of the ancestral strains.

## Increased mutation rates associated with the transition from carriage to disease

We found that the two disease isolates had accumulated single-base mutations at a faster rate (per site per generation) than the two carriage isolates, while there was no consistent difference in overall rate between isolates from the more and less pathogenic groups, or a correlation with genome size (Fig 1, Tables 1 and S3). The increased accumulation rate in disease isolates is observed across all classes of single-base mutations, including transitions, transversions, A/T to G/C transitions and transversions, and G/C to A/T transitions and transversions (Figs 2 and S3), and across both core and accessory genes (S4 Fig).

MA experiments aim to minimise the effect of selection on new mutations, so that accumulation rates reflect mutation rates as closely as possible. Our results indicate that the influence of selection on accumulation rates was low across our four MA experiments. First, distributions of single-base mutations across the 50 replicate lines do not differ significantly from a Poisson distribution for any of the four strains (S4 Table), suggesting that accumulation rates did not vary across lines. Second, rates of single-base mutation were similar across 1st and 2nd codon positions and 4-fold degenerate sites for each strain (S5 Fig). Third, non-unique single base mutations were rare: only 5 of the 2,267 single base changes observed over the four experiments occurred in more than one line, and none occurred in more than two. Fourth, we

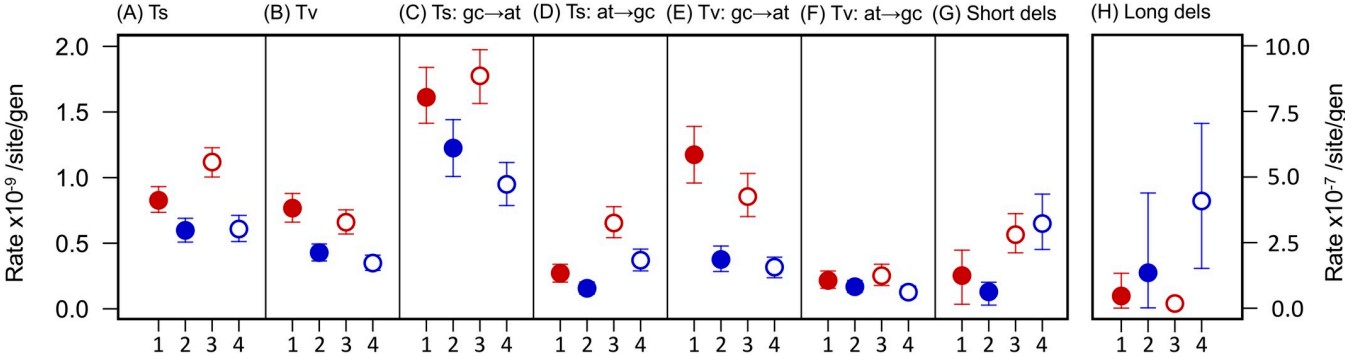

**Fig 2. Disease strains have faster rates of transitions (Ts) (A), transversions (Tv) (B), G/C to A/T transitions (C), A/T to G/C transitions (D), G/C to A/T transversions (E), and A/T to G/C transversions (F), but not deletions (G, H).** Disease strains are shown in red and carriage strains in blue. Strains from the more pathogenic group are shown as filled circles, and strains from the less pathogenic group are shown as empty circles. All points represent mean values across 50 replicate lines, and bars represent 95% confidence intervals estimated from bootstrapping across lines. Rates for (A)-(G) are shown on the left axis, and rates for (H) are shown on the right axis.

found no evidence of an overall change in maximum growth rates over the course of the experiment for the evolved lines of the two carriage strains or the disease strain with the smaller genome, and a net decline in maximum growth rate in the disease strain with the larger genome, consistent with its accumulation of the largest number of mutations, and these mutations tending to have a weakly deleterious effect (S6 and S7 Figs, and S5 Table). And finally, we sequenced five lines of each of the four strains at the midpoint of the experiment (100 days), and estimated rates of accumulation over the first and second halves of the experiment. This revealed no evidence of a change in rate (S6 Table).

To further test the observed difference in mutation rate between disease and carriage isolates we undertook an additional smaller scale (25-day) MA experiment with an additional four isolates. These four isolates were chosen to include much closer relatives of the two isolates from the more pathogenic group from our original experiment, and a more distantly related disease/carriage pair, also from the more pathogenic group (Tables 1, S1 and S7). This allowed us to investigate whether the difference between disease and carriage isolates holds over shorter evolutionary distances, and to test the generality of the association. 15 replicate lines of the four strains were evolved over 25 days, 11–13 randomly selected lines of each strain were sequenced, and mutations were identified in the same way as in the 200-day experiment.

Our estimates of single-base mutation rates for this combined data set of four pairs of disease and carriage isolates showed a consistent pattern: disease isolates have faster rates than closely related carriage isolates. Moreover, all disease isolates have faster rates than all carriage isolates. While there is uncertainty associated with the rate estimates for individual strains and our sample size is small, a two-sided paired t-test suggests that the difference between disease and carriage isolates we observe in our point estimates is unlikely to have arisen by chance: $p = 0.035$ (Fig 3 and S8 Table). This pattern appears to be independent of genome size as it holds across pairs with variable average genome sizes, and both across pairs in which the disease strain has a smaller genome than the carriage isolate (3 pairs) and a pair in which the disease isolate has a larger genome than the carriage isolate (Fig 3B and 3C). While we observe greater variation in overall mutation rates across the four carriage isolates in our two experiments than across the four disease isolates, we find no evidence that this variation is correlated with genome size (Figs 3 and S8).

## Changes in mutational spectrum associated with increased pathogenicity

Over the course of our 200-day MA experiment sufficient mutations accumulated to allow us to examine differences in mutational bias across isolates (S3 Table). Comparing the four isolates from the 200-day experiment, we found that while there is no consistent difference in the single-base mutation rate across isolates from more and less pathogenic groups there is a difference in the mutational spectrum. The two isolates from the more pathogenic group have a higher rate of transversions relative to transitions, and a higher rate of G/C to A/T transitions relative to A/T to G/C transitions (Fig 1F and 1G). In addition, while we observe a context-dependency of mutation rates (a higher mutation rate at sites flanked by a G/C base) in the two isolates from the less pathogenic group as in previous studies [29, 30], we do not observe this in the two isolates from the more pathogenic group (S9 Fig).

The differences in mutational bias across the different groups are observed consistently across both core and accessory genes (S10 Fig). Moreover, the rate of G/C to A/T mutations relative A/T to G/C mutations is correlated with the base composition of core genes (despite a small sample size, Pearson's correlation: $r^2 = 0.98$, $p = 0.01$; Fig 4). Comparison of our estimates of mutational bias and the base composition of core genes across these four strains also reveals that the two isolates from the more pathogenic group are further from their

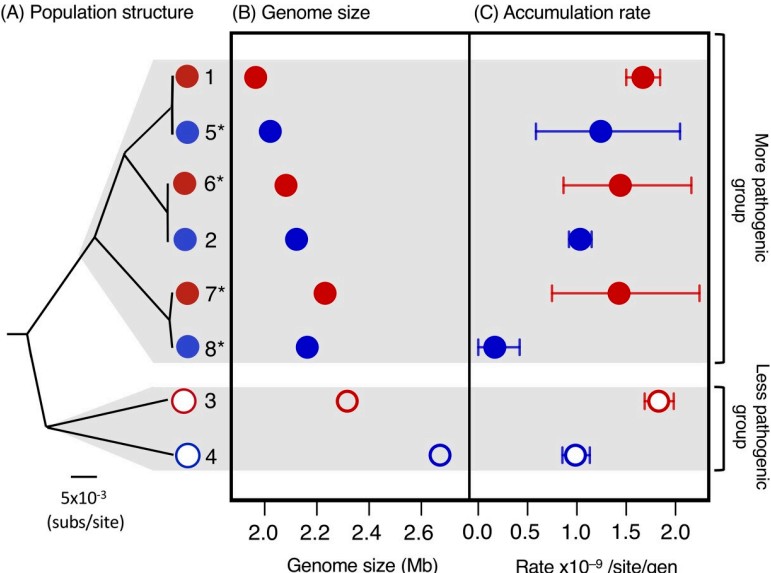

**Fig 3. Disease strains have faster mutation rates compared to closely related carriage strains, independent of genome size.** (A) The relationship between the eight strains in our two MA experiments based on a core gene alignment, with their location in the more and the less pathogenic groups in a phylogeny of a global collection indicated on the right (S1 Fig). (B) Genome sizes of the eight strains. (C) Point estimates of the genome-wide rate of single-base mutation, based on mean values across all replicate lines. Bars represent 95% confidence intervals estimated from bootstrapping across lines. Disease strains are shown in red and carriage strains in blue. Strains from the more pathogenic group are shown as filled points, and strains from the less pathogenic group are shown as empty points. A two-sided paired t-test suggests that the difference between disease and carriage isolates is unlikely to have arisen by chance: $p = 0.035$. Strain numbers match those in Fig 1 and Table 1, and isolates from the 25-day MA experiment are indicated with *.

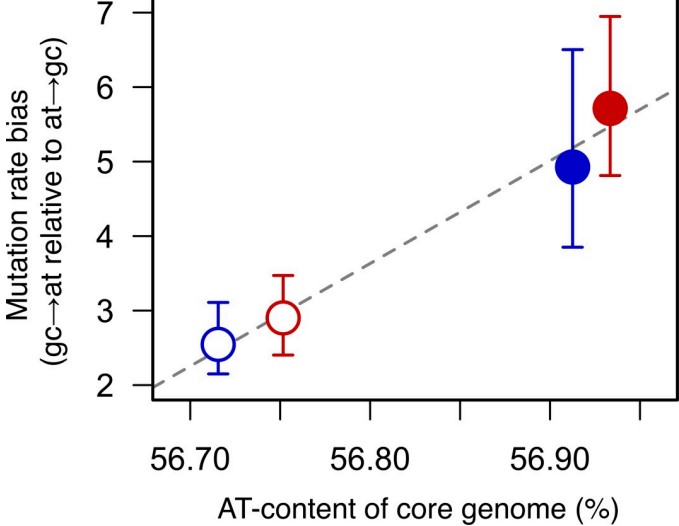

**Fig 4. Mutational bias may explain differences in core genome base composition.** The rate of accumulation of G/C to A/T mutations relative to the rate of accumulation of A/T to G/C mutations plotted against the proportion of bases that are A/T in genes common to all four strains from the 200-day MA experiment (1, 2, 3 and 4). All points represent mean values across 50 lines, and bars represent 95% confidence intervals estimated by bootstrapping across lines. Disease strains are shown in red and carriage strains in blue. Strains from the more pathogenic group are shown as filled circles, and strains from the less pathogenic group are shown as empty circles. The dashed line represents a line of best fit.

equilibrium base composition (S11 Fig). This suggests that the change in bias occurred in an ancestor of the two isolates from the pathogenic group, and may therefore be associated with their transition to a more pathogenic ecology.

## Deletion rates decline with genome reduction

Deletion mutations show a different pattern of variation to single-base mutations across the four strains in the 200-day MA experiment. When considering short insertions and deletions ($< 30$ bp), we observe a faster rate of deletion in the two isolates from the less pathogenic group, without a corresponding increase in the rate of insertion (Figs 2G and S12, and S9 Table). As most small deletions occur in intergenic regions, which tend to be shorter in the two isolates from the more pathogenic group, the difference may be due to the absence of intergenic regions that are more prone to deletion in these isolates.

In contrast, larger deletions ($> 100$ bp) arose at a faster rate in the two carriage isolates than the two disease isolates (particularly the carriage isolate from the less pathogenic group, Fig 2H). We identified 34 deletions larger than 100 bp across the four strains, with 10 larger than 10 kb (S10 Table). Most involved the loss of regions that show signatures of being mobile genetic elements (including phage-associated genes), which are more common in the isolates with larger genomes (S13 Fig, and S11 and S12 Tables).

## No loss of genes from DNA-repair pathways

We found no evidence that the loss or truncation of genes from known DNA repair pathways led to the observed differences in mutation rate or spectrum. Genes from the base excision repair, nucleotide excision repair and mismatch repair pathways were uniformly present across all eight strains (S14 Table). In addition, no genes that were consistently absent in disease strains were also consistently present in carriage strains (or vice versa) (S13 Table). Some DNA-repair pathway genes were present in multiple copies in some strains, but the presence/absence of these additional copies did not correlate with disease/carriage status or location in more/less pathogenic groups. In addition, the majority of DNA-repair pathway genes had a constant length across the eight strains, and where small length variations were observed, these did not correlate with disease/carriage status or presence in a more/less pathogenic group (S14 Table).

## Discussion

While we were only able to estimate the mutation rates of a small sample of *S. suis* isolates due to the time-consuming and labour-intensive nature of MA experiments, our results support two links between pathogenic ecologies and the rate and spectrum of mutations. First, *S. suis* isolates sampled from the site of invasive infections have consistently higher mutation rates than carriage isolates sampled from the tonsils. Second, isolates from the more pathogenic group of *S. suis* show a greater bias towards both G/C to A/T mutations and transversions, than those from a less pathogenic group. These changes in bias are uncoupled from disease/carriage status, and from overall mutation rate. In contrast, while genome size has previously been found to be a strong predictor of mutation rate variation across bacterial species (S8 Fig) [1, 16], we found no evidence of a link with mutation rates within *S. suis*. Disease isolates with a wide range of genome sizes have consistent mutation rates, and while the mutation rates of carriage isolates are more variable, they are not correlated with genome size. Nevertheless, as our sample size is small, further work is required to definitively test these associations.

In opportunistic pathogens, such as *S. suis*, carriage is thought to be a prerequisite of invasive disease, with carriage isolates acting as a source of disease isolates [31]. This transition

from asymptomatic carriage to invasive disease is likely to involve both growth in novel environments, and additional pressures from host immune systems [4]. Under these conditions, isolates with faster mutation rates may have an indirect selective advantage because they are more likely to generate novel adaptive variants [2, 3, 22], or because mutation rate is linked to another trait that is adaptive in these new conditions [32, 33]. While the strains in our experiments only represent a single isolate from any host, the difference we observe between carriage and disease isolates plausibly reflects within-host evolution. If this is the case, the rapidity of the change in rates suggests that it is driven by positive selection rather than random drift. This might involve selection on standing variation in carriage populations; variants with higher mutation rates could be present at low frequencies in carriage populations and rise to high frequencies in invasive populations. This is consistent with the results of previous studies that have identified variation in mutation rates in within-host bacterial populations [6, 18, 21, 23, 34], and with our observation of no correlation between the difference in rate and the evolutionary distance between disease and carriage isolates.

In contrast to previous studies that have identified hypermutable strains, the rate variation we observe between disease and carriage isolates is not associated with the loss or pseudogenisation of genes in DNA-repair pathways, and is much smaller in scale. While this means a smaller increase in the frequency of adaptive mutations, it also means a smaller increase in the frequency of deleterious mutations. These higher rates may therefore be maintained for longer and reach higher frequencies in a population than larger rate increases. This could contribute to the maintenance of mutation rate heterogeneity within populations, which is predicted to promote faster responses to selective challenges, while reducing the long-term burden of deleterious mutations [7, 35].

In addition to the difference in point mutation rates across disease and carriage isolates, we observed a difference in deletion rates. While we only have the power to estimate rates of deletion in our 200-day MA experiment, almost all of the large deletion events in this experiment occurred in the two carriage isolates, and often involved the loss of regions that contain phage-associated genes. This difference in rates might reflect fewer temperate prophages in the genomes of disease isolates. In *S. suis* the transition from carriage to disease is associated with a reduction in genome size, and it has been suggested that this is, in part, due to the loss of mobile genetic elements [12]. While the rate of prophage loss observed in our experiments is probably too slow to explain genome reduction during the transition from carriage to disease within a host, the stress associated with this transition might lead to a higher rate of loss [36].

Previous studies have found evidence that several clusters of *S. suis* have evolved to become more pathogenic to pigs, and are also responsible for zoonotic disease in humans [12, 28]. The emergence of these more pathogenic clusters has previously been shown to be associated with both genome reduction and an increased AT-richness of the core genome [12]. Here, we have found evidence that it is also associated with a change in mutational spectrum. While this is only supported by comparison of the four strains from our 200-day experiment (due to too few mutations in the 25-day MA experiment) our observation of a strong correlation between mutational bias and core genome base composition, is consistent with this underlying a broader association between AT-richness and pathogenicity in *S. suis* that was identified in a previous study [12]. Increased AT-richness is a common correlate of genome reduction in bacterial symbionts [8, 9, 37], and has also been observed in the evolution of the pathogen *Shigella* [38]. As G/C to A/T mutations tend to be more deleterious than A/T to G/C mutations, these patterns are commonly attributed to increased genetic drift [8, 9]. While our results are consistent with this as an ultimate explanation of the increased AT-richness in more pathogenic clusters of *S. suis*, the correlation we observe between mutational bias and core genome nucleotide composition suggests that it is an immediate consequence of a change in mutational bias. In

addition, our observation that the change in bias is not accompanied by an increase in overall rate suggests that it is unlikely to lead to a substantial increase in the burden of deleterious mutations [39, 40].

Overall, our results indicate that mutation rate variation within bacterial species extends beyond hypermutable strains, and can be sensitive to ecological transitions. Both the changes in overall rate and in mutational bias we observe in *S. suis* could influence the frequency of adaptive mutations, and therefore the capacity to rapidly respond to selective challenges such as evading host immunity or antibiotic treatments [41, 42]. This link between ecology and mutation rate could therefore prove to be both important in understanding the evolutionary trajectories of bacterial pathogens, and a useful marker of ecological change.

## Methods

### Strains and culture conditions

Strains were selected from a collection of *S. suis* isolates described in [12] together with three isolates sampled from pig farms in Europe (S2 Table). Isolates were sampled from different pig farms in Canada, Spain, the Netherlands and the UK, between 2009 and 2017. Isolates were classified as "disease" if they were recovered from systemic sites in pigs or humans with clinical signs consistent with *S. suis* infection, or from the lungs of a pig with signs of pneumonia. Isolates recovered from the tonsils or tracheo-bronchus of healthy pigs or pigs without signs of *S. suis* infection were classified as "carriage". Strains for both experiments were chosen based on clinical information, genome size, and location in a core genome phylogeny previously described in [12]. All strains were from pigs, and all "disease" isolates were associated with systemic infections.

The solid media used for all experiments was Todd-Hewitt broth (THB) supplemented with 1.5% (w/v) agarose and 0.2% yeast extract (THY) (Oxoid, Basingstoke). 20% Glycerol/THB (36.4g in 1 litre) with 0.2% yeast extract was used for archiving and PCR, and THB with 0.2% yeast extract was used for overnight growth and growth rate experiments. Cultures were streaked to single colonies on solid media and incubated overnight in a static incubator at 37˚C to generate stock plates. For overnight cultures, an independent colony was picked from the stock plate to inoculate THB + 0.2% yeast extract broth and incubated in a static incubator at 37˚C.

### Mutation accumulation experiments

75 replicate lines were established from each strain in the 200-day mutation accumulation (MA) experiment and 15 replicate lines in the 25-day MA experiment. Each line was passaged through single-colony bottlenecks by selecting the last visible independent colony in the streak (to minimise selection bias). Plates were incubated at 37˚C for approximately 24 hours between each transfer. All passaged plates were stored at 4˚C for 24 hours, to allow lines that failed to grow during passage to be reset from the stored plate. In the 200-day experiment lines were archived in liquid broth every 14 days, and every 100 days.

### PCR

For the 200-day experiment, the four strains were grown on quadrants of a single plate. To allow for errors during passaging to be corrected we developed a multiplex PCR to distinguish the strains. Two sets of primer sequences for a multiplex PCR were designed using a custom Python script to identify unique regions of each strain of set lengths that would resolve in electrophoresis, along with a *S. suis* positive control (S15 Table). To conduct the PCR, a single

colony was transferred to 150 μl liquid media, 5μl of which was added to 50 μl sterile MiliQ water and heated at 95˚C for six minutes. Amplification conditions and reagent volumes are given in S15 Table. We tested all lines every 14 days. If a mismatch was identified, lines were set back to the previous PCR run.

### Estimation of generation times

Strains were streaked to single colonies from overnight cultures on THY agar plates and incubated at 37˚C for 24 hours, with a minimum of three biological replicates. Single colonies were collected by excising a small disc of agar around a colony and resuspending it in 10 ml of PBS. The solution was serially diluted, and dilutions spread on THY agar plates. The plates were incubated in a static incubator at 37˚C and examined after 24 hours to determine the number of colony forming units (CFU). Generation times were calculated by dividing $\log_2$(average CFU) by the period of growth (S2 Table and S2 Fig).

### Growth rate measurements

Growth rates were estimated for the ancestral lines and a random sample of 25 of the sequenced evolved lines of each strain in the 200-day experiment, with a minimum of three biological replicates. Overnight cultures were pelleted by centrifugation at 4000xg for 3 minutes to remove spent media. After discarding the supernatant, the cell pellet was resuspended in fresh THB +0.2% yeast extract media to a final concentration of $10^7$ CFU per well. 300μl of the culture was transferred into wells and incubated in a Bioscreen C (Oy Growth Curves Ab Ltd) at 37˚C, with optical density ($OD_{600}$) measured every 5 minutes for 24 hours. For 10 evolved lines (9 of strain 1 and one of strain 3) there was insufficient overnight growth to reach the required starting concentration, and growth rates could not be accurately measured. Maximum growth rates were calculated by taking the slope of a linear regression model of $\log_2(OD_{600})$ over time, using a 30-minute sliding window to identify the period of fastest growth (for details, see [43]). To test the reliability of OD as a proxy for CFU during exponential growth, additional time-sampled growth curves were completed, measuring both OD and CFU for each of the four ancestral strains (S16 Table).

### Sequencing

Illumina whole genome sequencing was undertaken for all ancestral strains, a random sample of 50 evolved lines of each of the strains in the 200-day MA experiments, 5 evolved lines of each of the strains in the 200-day MA experiments at day 100, and 11–13 evolved lines of each of the strains in the 25-day MA experiments. DNA extraction, library preparation and sequencing using a HiSeq 2500 instrument (Illumina, San Diego, CA, USA) was undertaken by MicrobesNG (Birmingham, UK).

### Long-read sequencing, assembly and annotation of ancestral strains

We assembled high-quality reference genomes for all eight ancestral strains using methods that combine short-read and long-read sequence data. For the 200-day experiments, long-read sequencing library preparation was performed using Genomic-tips and a Genomic Blood and Cell Culture DNA Midi kit (Qiagen, Hilden, Germany). Sequencing was performed on the Sequel instrument from Pacific Biosciences using v2.1 chemistry and a multiplexed sample preparation. Reads were demultiplexed using Lima in the SMRT link software (https://github.com/PacificBiosciences/barcoding). Reads shorter than 2500 bases were removed using prinseq-lite.pl (https://sourceforge.net/projects/prinseq/). Hybrid assemblies, using filtered PacBio

and Illumina reads, and preliminary assemblies of long-read data generated with Canu v1.9 [44] were generated with Unicycler v0.4.7 using the normal mode and default settings [45]. Assembly graphs were visualised and, if necessary, manually corrected with Bandage [46]. For the 25-day MA experiments, library preparation, short-read and long-read sequencing, and hybrid assembly with Unicycler were undertaken by MicrobesNG as part of their Enhanced Genome Service, which uses both Illumina and Oxford Nanopore Technologies. For 4/8 strains the assemblies were single-contig, and for the other four the longest contig was >98% of the total assembly length (smaller contigs likely representing plasmids or mobile elements).

Genomes were annotated using Prokka v1.14.5 [47] and putative mobile elements identified using IslandViewer 4 [48]. Panaroo v1.2.2 [49] was used to identify orthologous genes (using recommended parameter settings) and create alignments of shared genes. Core-genome distance matrices were estimated and a neighbour-joining tree created using these alignments and the *ape* package in *R* [50].

## Mutation calling in evolved lines

Mutations in evolved lines were called by mapping short-read sequence data to the reference genomes of the ancestral strains. Illumina reads were adapter-trimmed using Trimmomatic with a sliding window quality cutoff of Q15 [51]. They were mapped to the ancestral strain (excluding any short contigs) using Bowtie2, and variants called using SAMtools and BCFtools [52, 53]. False variant calls were identified using several approaches. First, we excluded any calls with a depth of less than six reads (average coverage was always >30x) or where the reference allele was present in more than 5% of reads. Second, short-read data from the ancestral strains was mapped back to itself, and any variants identified were excluded. Finally, any variant calls that either had a lower quality score than the maximum for the line, were within 100 bases of another variant call in any line, or were within 1000 bases of the start or end of the reference genome, were checked by eye for evidence of mapping error. In the 200-day experiment, clusters of mutations were identified (mutations within 30 bases of another in the same line). They were common only in strain 4 (<5% of mutations in the other three strains), and only 10% fell in regions of the genome shared across the 4 strains. These mutations were not excluded from our core analyses, but S4 and S5 Figs describe the impact of their exclusion.

Indels were identified using both SAMtools and ScanIndel [53, 54]. To avoid false calls, we required indels to be identified by both analyses and not identified when short-read data for the ancestral strain was mapped back to the reference assembly. In addition, we required that the alternate allele is supported by at least five reads, at least one in each orientation, and fewer than 5% of reads supporting the reference allele, a quality score in SAMtools of at least 20, and an 'IMF' of at least 0.8. Deletions longer than 100 bases were called by identifying extended regions with zero coverage in our mapped assemblies, and confirmed through examination by eye.

## Estimating average rates and statistical comparisons

The single-base mutation rate per site per generation for each strain was estimated as:

$$\mu_{sb} = \frac{m}{ng}$$

where *m* is the number of observed single-base mutations, *n* is the number of nucleotide sites analysed (genome length multiplied by the number of lines), and *g* is the total number of generations over the duration of the experiment (see above). This value was estimated for the whole genome, for different subsets of sites, and for different types of single base mutations,

with the number of nucleotide sites analysed adjusted in each case. 95% confidence intervals were generated for estimates using two approaches. First, based on 1000 bootstrap samples of the lines of each strain, and second based on estimates of the standard deviation in the rate across lines. We found that confidence intervals were similar across the two methods, and none of our results were contingent on the use of either method.

### Identification of DNA-repair genes

DNA-repair genes were identified from the Prokka and Panaroo annotations using descriptions of genes in the *S. suis* DNA mismatch repair, base excision repair and nucleotide excision repair pathways from KEGG [55].

### Supporting information

**S1 Fig. Population structure of a global collection of *Streptococcus suis* isolates.** (a) A core genome phylogeny of 962 isolates of *S. suis* [12]. Individual disease (red) and carriage (blue) isolates are indicated in the inner strip. The more pathogenic clade is indicated by a red outer ring, and the less pathogenic clade by a blue outer ring. The locations of each strain in our two MA experiments are indicated on this strip (Table 1). (b) The proportion of isolates in each clade that are associated with disease (excluding isolates for which disease-association is unknown). (c) A box plot of genome sizes of isolates from each clade. (d) A box plot of the core genome GC-content of isolates from each clade.
(PDF)

**S2 Fig. Estimates of number of generations per day based on colony counts for each ancestral strain.** Points represent mean estimates of the number of colony forming units (CFU) present after 24 hours of growth from a single CFU for each ancestral strain, on a $\log_2$ scale. Bars show the range of values returned across biological replicates (at least 3 biological replicates of each strain). Disease strains are shown in red and carriage strains in blue. $\log_2$(CFU) after 24 hours of growth gives an estimate of the number of generations over that period. We find no evidence of a difference in generation time between disease and carriage strains, but the two strains from the less pathogenic clade (strains 3 and 4) have a longer generation time than the six strains from the more pathogenic clade (strains 1, 2, 5, 6, 7 and 8).
(PDF)

**S3 Fig. Comparison of mutation rates for different types of single nucleotide variants in the 200-day MA experiment.** Points represent mean values across 50 replicate lines, and bars represent 95% confidence intervals estimated from bootstrapping across lines. Numbers relate to Table 1; disease strains are shown in red and carriage in blue, strains from the more pathogenic clade are shown as filled shapes and strains from the less pathogenic clade as empty shapes.
(PDF)

**S4 Fig. Comparison of mutation rates across core and accessory genes in the 200-day MA experiment, with (a) and without clustered mutations (b).** Estimates of rates of single-base mutation rates across accessory (squares) and core (circles) genes. Points represent mean values across 50 replicate lines, and bars represent 95% confidence intervals estimated from bootstrapping across lines. Numbers relate to Table 1; disease strains are shown in red and carriage in blue, strains from the more pathogenic clade are shown as filled shapes and strains from the less pathogenic clade as empty shapes.
(PDF)

**S5 Fig. Comparison of mutation rates in the 200-day MA experiment across different categories of sites, with (a) and without clustered mutations (b).** Both figures show rates for 1st/2nd codon positions, four-fold degenerate sites, and intergenic sites, for both regions that are shared across all four strains and regions that are not shared. Shared intergenic sites were defined as intergenic regions between genes that were syntenic across all four strains. Estimates based only on genes that are shared across all four strains (core) are shown as circles, and estimates based only on genes that are not shared across all four strains (accessory) are shown as squares. Disease strains are shown in red and carriage strains in blue. All points represent mean values across 50 replicate lines, and bars represent 95% confidence intervals estimated from bootstrapping across lines.
(PDF)

**S6 Fig. Comparison of the maximum growth rate estimates for evolved and ancestral lines of the four strains in the 200-day MA experiments.** Histograms of maximum growth rates (change in optical density (OD) per min) for evolved lines (mean values across 3 biological replicates; shown in grey) and repeated measurements of the ancestral line (red for disease and blue for carriage). Estimates were attempted for a random sample of 25 evolved lines for each strain, however insufficient overnight growth meant that we could not obtain accurate maximum growth rate estimates for 9/25 evolved lines of strain 1 and 1/25 evolved lines of strain 2. Vertical lines represent mean values across repeated measurements of the ancestral strain (red/blue), and across evolved lines (grey). For strain 3 there is evidence of a net decline in maximum growth rate in the evolved lines (Welch's t-test, $p = 2.9 \times 10^{-5}$, indicated by * above brackets). For the three other strains there is no evidence of a net change in the maximum growth rate. Estimates of maximum growth rate for 7 individual evolved lines were significantly lower than estimates for the ancestral strain, and 1 significantly higher (Welch's t-test, $p < 0.05$ after Bonferroni correction for multiple testing, indicated by * above bars).
(PDF)

**S7 Fig. Relationship between maximum growth rate and numbers of single-base mutations accumulated in the evolved lines of the four strains in the 200-day MA experiments.** Maximum growth rate (change in optical density (OD) per min) plotted against the number of single-base substitutions accumulated over the course of the experiment for evolved lines of each strain. Growth rate estimates were attempted for a random sample of 25 evolved lines of each strain, however insufficient overnight growth meant that we could not obtain accurate maximum growth rate estimates for 9/25 lines of strain 1 and 1/25 lines of strain 2. The estimates of maximum growth rates shown are mean values across 3 biological replicates. Spearman's $\rho$ and p-values for correlation tests are reported for each strain. There is no evidence of a significant correlation between maximum growth rate and the number of mutations accumulated for any of our four strains. However, the 9 lines of strain 1 that had insufficient overnight growth had more single-base mutations (average of 15.4) than 16 lines that had sufficient overnight growth (average of 11.4) (Welch's t-test, $p = 0.04$). Similarly, the line of strain 2 than had insufficient overnight growth had a high number of single-base mutations (n = 13) compared to the other lines of this strain included in the experiment.
(PDF)

**S8 Fig. (a) Mutation rate estimates for *S. suis* in the context of those of other species, and (b) a comparison of mutation rate estimates per genome.** (a) Mutation rate estimates against genome size, for *S. suis*, *Mesoplasma florum*, *Helicobacter pylori*, *Thermus thermophilus*, *Staphylococcus epidermidis*, *Deinococcus radiodurans*, *Vibrio cholerae*, *Vibrio fischeri*, *Bacillus subtilis*, *Mycobacterium tuberculosis*, *Escherichia coli*, *Salmonella typhimurium*, *Salmonella enterica*,

*Teredinibacter turnerae*, *Agrobacterium tumefaciens*, *Pseudomonas aeruginosa*, *Mycobacterium smegmatis* and *Burkholderia cenocepacia* [1, 56]. *S. suis* disease isolates are shown in red and carriage isolates in blue, with closed circles representing isolates from the more pathogenic clade and open circles isolates from a less pathogenic clade, other species are shown in grey. In (b) the dashed line indicates the value of the constant mutation rate identified by Drake in his original study of the relationship between mutation rate and genome size [16].
(PDF)

**S9 Fig. The influence of flanking sites on mutation rates for the four strains in the 200-day experiment.** Estimates of mutation rates for sites that have at least one G/C flanking site, and estimates of rates for sites that have no G/C flanking site for each of the four strains from the longer experiment. Points represent mean values across 50 lines, and bars represent 95% confidence intervals from bootstrapping across lines.
(PDF)

**S10 Fig. Mutational biases in shared and non-shared genomic regions in the four strains in the 200-day experiment.** (a) The ratio of transitions to transversions for shared (circles) and non-shared (squares) regions of the genome. (b) The ratio of G/C to AT transitions to A/T to G/C transitions for non-shared (squares) and shared (circles) regions of the genome. Disease-associated strains are coloured in red circles and carriage strains in blue, with filled shapes representing isolates from the more pathogenic group and empty shapes isolates from the less pathogenic group. All points represent mean values across 50 lines, and bars represent 95% confidence intervals estimated by bootstrapping.
(PDF)

**S11 Fig. Equilibrium GC-content for the four strains in the 200-day MA experiment.** Equilibrium GC-content estimates, calculated from the rates of A/T to G/C mutation and G/C to A/T mutation (circles) and actual genome-wide GC-content for the four strains (squares). Disease-associated strains are coloured in red and carriage strains in blue, with filled shapes representing isolates from the more pathogenic group and empty shapes isolates from the less pathogenic group. All points represent mean values across 50 lines, and bars represent 95% confidence intervals estimated by bootstrapping.
(PDF)

**S12 Fig. Rates of accumulation of small indels for the 200-day MA experiment.** (a) The rates of loss/gain of nucleotide bases through short deletion (circles) and insertion (square) events, and (b) the rates of short deletion (circle) and insertion (squares) events. All points represent mean values across 50 lines, and bars represent 95% confidence intervals estimated by bootstrapping.
(PDF)

**S13 Fig. Length of elements identified as mobile genomic islands by IslandViewer for the four strains in the 200-day MA experiment.** Figures show the relationship between the total length of mobile genetic elements in each strain and total genome size (a), and the relationship between the length of the genome excluding mobile genetic elements and total genome size (b).
(PDF)

**S1 Table. Description of the eight strains used in both MA experiments.**
(DOCX)

**S2 Table. Counts of colony forming units (CFUs) after 24 hours of growth for the 8 ancestral strains used to estimate generation time for each strain.**
(XLSX)

**S3 Table. Description of all single-base mutations observed for each of the 50 lines of the 4 strains in the 200-day experiment.** The table describes the position of the mutations in the (single contig) assembly of the ancestral strain, the inferred single base change, the category of site, whether or not the site is shared across strains ('Shared region'), whether the site is on the leading or lagging strand, the two flanking bases, and whether or not the mutation is clustered (within 30 bp of another mutation in that line). The site categories are 1st/2nd codon positions (1), 3rd codon positions (3), 4-fold degenerate sites (4) and intergenic (0). Shared sites (1) are defined by whether or not the gene is present in all strains for sites within genes, and whether or not the two flanking genes are present in all strains for intergenic regions.
(XLSX)

**S4 Table. Results of Chi-Squared test for the goodness of fit of a Poisson distribution to mutation rates of single-base substitutions in the 200-day MA experiment.** Counts were divided into four categories for each strain, with each category having an expected frequency >5 under a Poisson distribution given our estimates of the mean rate.
(DOCX)

**S5 Table. Growth rates for ancestral and evolved lines in the 200-day MA experiment.**
(XLSX)

**S6 Table. Comparison of numbers of single-base mutations observed at day 100 and day 200 in the 200-day experiment.** 5 lines of each strain in the 200-day MA experiment were sequenced at the mid-point of the experiment to establish whether faster rates were transitory. Lines were selected that had higher than average rates from each strain. We found no evidence of a difference in rate over the first and second half of the experiment.
(DOCX)

**S7 Table. Core genome pairwise nucleotide distances between the eight ancestral strains used in the MA experiments.** The proportion of nucleotide bases that differ between pairs of strains in the MA experiments based on an alignment of shared genes generated by Panaroo. Closely related disease/carriage pairs are highlighted.
(DOCX)

**S8 Table. Description of all single-base mutations observed for each of the lines of the 4 strains in the shorter experiment.**
(XLSX)

**S9 Table. Description of all short indel mutations observed for each of the 50 lines of the 4 strains.**
(XLSX)

**S10 Table. Description of all long deletion mutations observed for each of the 50 lines of the four strains.** Each row is a line in which a long deletion was identified. Lines in which no long deletion was identified are not described. The locations and lengths of the deletions are described, and genes in the deleted regions described. Whether or not the deleted region was identified as a mobile genetic element (MGE) by IslandViewer is described (1 = yes).
(XLSX)

**S11 Table. The size of regions identified as mobile genetic elements for each strain in the 200-day MA experiment.** The lengths of the regions identified by IslandViewer for each strain.
(DOCX)

**S12 Table. The regions identified as mobile genetic elements by IslandViewer.**
(XLSX)

**S13 Table. Table of gene presence/absence across 8 strains.** The output of Panaroo for the eight strains used in our MA experiments.
(XLSX)

**S14 Table. Genes in major DNA repair pathways in the 8 strains.**
(XLSX)

**S15 Table. Oligonucleotide primers (Sigma-Aldrich) and conditions used for Multiplex PCR strain identification.**
(DOCX)

**S16 Table. The relationship between CFU count and OD during exponential growth in the four ancestral strains from the 200-day experiment.** The relationship between OD and CFU count can vary across bacterial strains. To test whether OD is a reliable indicator of CFU count in the strains in our 200-day experiment, we undertook an additional growth rate experiment following the same procedure as before. Every half hour for the first 5 hours of growth, then every hour up until 8 hours of growth, and at the end of the experiment (24 hours of growth), 100µl of the 300µl culture was removed from a single well. This was serially diluted and spot plated on three THY plates. Comparisons of OD and CFU count revealed a linear relationship in all four ancestral strains during the period of exponential growth (between 0.5 and 3.5 hours). For all four strains OD was found to be a good predictor of change in CFU count during exponential growth (Pearson's correlation coefficient >0.9 for each strain).
(XLSX)

## Acknowledgments

Genome sequencing was undertaken by MicrobesNG (http://www.microbesng.uk/), which is supported by the BBSRC (grant number BB/L024209/1), and by the Sanger Institute in collaboration with Julian Parkhill. We wish to thank John Welch and Olivier Restif for helpful comments on the manuscript, and Virginia Aragon and her team for one of our strains.

## Author Contributions

**Conceptualization:** Gemma G. R. Murray, Nazreen F. Hadjirin, Eric Miller, Lucy A. Weinert.

**Data curation:** Gemma G. R. Murray, Sebastian Bruchmann, Eric Miller.

**Formal analysis:** Gemma G. R. Murray, Andrew J. Balmer, Josephine Herbert, Sebastian Bruchmann.

**Funding acquisition:** Eric Miller, Lucy A. Weinert.

**Investigation:** Gemma G. R. Murray, Andrew J. Balmer, Josephine Herbert, Nazreen F. Hadjirin, Caroline L. Kemp, Marta Matuszewska, A. S. Md. Mukarram Hossain, Eric Miller, Lucy A. Weinert.

**Methodology:** Gemma G. R. Murray, Eric Miller, Lucy A. Weinert.

**Project administration:** Eric Miller, Lucy A. Weinert.

**Resources:** Marcelo Gottschalk.

**Supervision:** Gemma G. R. Murray, Nazreen F. Hadjirin, Marta Matuszewska, Alexander W. Tucker, Eric Miller, Lucy A. Weinert.

**Visualization:** Gemma G. R. Murray.

**Writing – original draft:** Gemma G. R. Murray.

**Writing – review & editing:** Gemma G. R. Murray, Andrew J. Balmer, Josephine Herbert, Lucy A. Weinert.

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
