## [Decision Letter · Decision Letter 0]

30 Jul 2021

Dear Dr Murray,

Thank you very much for submitting your Research Article entitled 'Mutation rate dynamics reflect ecological change in an emerging zoonotic pathogen' to PLOS Genetics.

The manuscript was fully evaluated at the editorial level and by three independent peer reviewers. The reviewers appreciated the attention to an important problem, but raised some substantial concerns about the current manuscript. Based on the reviews, we will not be able to accept this version of the manuscript, but we would be willing to review a much-revised version. We cannot, of course, promise publication at that time.

If you decide to revise the manuscript for further consideration at PLOS Genetics, please aim to resubmit within the next 60 days, unless it will take extra time to address the concerns of the reviewers, in which case we would appreciate an expected resubmission date by email to plosgenetics@plos.org.

[LINK]

We are sorry that we cannot be more positive about your manuscript at this stage. Please do not hesitate to contact us if you have any concerns or questions.

Yours sincerely,

Jianzhi Zhang

Associate Editor

PLOS Genetics

Gregory P. Copenhaver

Editor-in-Chief

PLOS Genetics

Reviewer's Responses to Questions

**Comments to the Authors:**

Reviewer #1: The authors quantitatively measure mutation rate in 4 pairs of S. suis strains. One member of each pair is pathogenic, while the other is commensal. In each of the 4 pairs, the pathogenic clone has a higher mutation rate than the commensal clone. From these findings, the authors suggest that "ecology is a stronger correlate of mutation rate than genome size over these timescales, and that transitions to invasive disease are consistently accompanied by rapid increases

in mutation rate".

I believe the rigor of the authors' protocol for measuring mutation rates. However, my opinion is that two aspects of the authors' claims are too strong with respect to what their data show. I suggest that the authors either temper these claims, or provide more conclusive evidence to support them.

These claims are: 1) recent pathogenic strains have higher mutation rates than commensal carriage strains of S. suis. 2) transitions to invasive disease are consistently accompanied by rapid increases in mutation rate.

Regarding claim 1. While the authors' data is suggestive, I do not think we can draw a definitive conclusion about the association between ecology and mutation rate in S. suis. The small sample size (4 pairs of clones-- due to phylogenetic correlations between all clones, I see this as 4 independent data points) prevents a strong conclusion in this regard. As a quick statistical sanity check, the odds of the pathogen strain having a higher mutation rate than the commensal strain in all four pairs by chance, is the same as the odds of flipping a coin 4 times and getting 4 heads (p = 0.065). Hence, I am not convinced that the 4 pairs of recent pathogen/commensal strains is sufficient to claim a strong association between ecology and mutation rate in S. suis. I would prefer to see more pairs so that the authors can make definitive statistical claims for or against their hypothesis. Alternatively, the authors could weaken their conclusions, by stating that these data favor their hypothesis, but that more data (more pathogen/commensal pairs-- not more MA lines from the existing pairs) are needed to definitively test it.

Regarding claim 2. The authors assume throughout that the pathogen strain recently evolved from the commensal strain, and than a higher mutation rate evolved from a lower mutation rate. However, from the data shown, the converse scenario is equally likely. That is, the commensal strain could have evolved from the pathogen, and a lower mutation rate could have evolved from a transition to a non-pathogenic lifestyle. This possibility is never discussed in the manuscript. Perhaps the authors have data or reasons for excluding this possibility, but these reasons are not clearly stated.

Reviewer #2: In this paper, the authors set out to test one question, if the invasive disease isolates of the pathogen S. suis have higher mutation rate than their asymptomatic relatives? The reasoning behind this is that the living environment for the invasive disease isolates is more stressful and thus may benefit from a higher mutation rate.

To test this idea, the authors first did MA experiment in four isolates of S. suis, two invasive and two asymptomatic. 75 replicates were established for each isolate and the experiment went on for 200 days (close to 4000 generations). This experiment allows them to determine the mutation rate of these four isolates with great accuracy. And they found, in both pairs of comparison, the invasive disease isolate has significantly higher mutation rates than their asymptomatic isolate, the differences are close to two folds.

The first part of the results looks promising, but it is difficult to draw conclusion with just two pairs of comparison, the authors then did a smaller scale MA experiment in another four isolates (15 replicates each isolate, 25 days, and round 500 generations). The results from these two experiments agree with each other. Now there is significant difference in mutation rate between invasive disease isolates and asymptomatic isolates as shown by a paired t-test (P=0.03).

Apart from the major conclusion, the authors also found that there are differences in mutation spectrum between invasive disease isolates and asymptomatic isolates, and a correlation between mutational AT bias with genomic GC content.

I find the analysis in this paper robust and results supportive, and I recommend acceptance with a minor revision.

My questions are listed below:

1. Data availability

(1) The sequence reads should be made publicly available;

(2) The assembled genomes should also be made publicly available;

2. Sample information in the first Methods section. The authors cited another paper on the origination of the samples, it might be better to provide the information in this paper, like which part of the world are they collected and are some of them are from the same farm.

3. Lines 172-173. It might be interesting to look at the spectrum of all SNV mutations (relative frequency of 6 types of SNVs). Examples like this can be found in papers that measure mutation rate.

4. The scale bar of the phylogenetic tree in Fig 3 is missing.

Reviewer #3: Despite its strengths, the initial premise of the manuscript in the abstract is incorrect because mutation rates evolve as linked traits, not as traits directly under selection. While this is a short logical walk for asexual, infrequently recombining haploids, it’s still important to clarify that selection does not “choose” genotypes on the basis of their mutation rates but rather on the side effects of these rates.

We also know a great deal about why mutation rates vary so much in the first place – it’s because of varying effective population sizes, after work by Lynch, Keightley, Foster and many others. Small population sizes enhance effects of drift relative to selection, and this weakened purifying selection allows mutations that erode replication fidelity to become frequent and even fix. So the primary hypothesis of this study should be that pathogenic lineages have smaller Ne (which seems very logical based on their newly limited lifestyle) and consequently their mutation rates increase.

The primary results, higher mutation rates and spectra consistent with MMR defects (GC->AT) in pathogenic Strep. suis, make great sense. There’s limited statistical power in these comparisons unfortunately, even though the MA experiments involve a lot of well-done work that deserve celebration. There are only ~2200 mutations, or <300 per strain, and while this is a comparable amount to other well-regarded MA studies, it still makes comparisons between strains that differ in several, inconsistent ways (over relatively small evolutionary distances between them) challenging.

Still, the fact that isolates from invasive infections had somewhat (not quite twofold?) higher mutation rates than carriage isolates is intriguing, and the negative result showing no correlation between substitution rate and genome size is useful.

The observation of fewer deletions in smaller genomes is interesting and a novel finding as far as I know, but I’m curious how many indels you actually reliably detected? Indels are usually about ten times rarer than SNPs, meaning that the total observations may again be limiting.

To summarize, I really like the premise and effort of this study, and the results are valuable in general, but the causal relationships demand revision and the limited power of the cross-strain comparisons need to be acknowledged and explained.

**Have all data underlying the figures and results presented in the manuscript been provided?**

Reviewer #1: Yes

Reviewer #2: **No: **Sequencing reads are not provided.

Reviewer #3: Yes

PLOS authors have the option to publish the peer review history of their article (what does this mean?). If published, this will include your full peer review and any attached files.

Reviewer #1: **Yes: **Rohan Maddamsetti

Reviewer #2: No

Reviewer #3: No

---

## [Decision Letter · Decision Letter 1]

6 Oct 2021

Dear Dr Murray,

We are pleased to inform you that your manuscript entitled "Mutation rate dynamics reflect ecological change in an emerging zoonotic pathogen" has been editorially accepted for publication in PLOS Genetics. Congratulations!

Please note that Reviewer #1 has some minor suggestions (see below) which you can consider as you prepare your final draft fore the production team (the editorial team will not need to re-evaluate). Before your submission can be formally accepted and sent to production you will need to complete our formatting changes, which you will receive in a follow up email. Please be aware that it may take several days for you to receive this email; during this time no action is required by you. Please note: the accept date on your published article will reflect the date of this provisional acceptance, but your manuscript will not be scheduled for publication until the required changes have been made.

Yours sincerely,

Jianzhi Zhang

Associate Editor

PLOS Genetics

Gregory P. Copenhaver

Editor-in-Chief

PLOS Genetics

Comments from the reviewers (if applicable):

Reviewer's Responses to Questions

**Comments to the Authors:**

Reviewer #1: I appreciate the authors' care in revising their manuscript. I find the presentation much improved. Even though these data may not offer definitive answers due to small sample size, they are certainly very interesting.

I strongly recommend striking the first sentence of the abstract, and starting with the second sentence "Mutation rates vary both within ...".

The first sentence lacks nuance, and associates this work with longstanding controversies in the field (https://en.wikipedia.org/wiki/Adaptive_mutation). I do think the discussion of mutation rate variation in the rest of the paper is quite good, both measured and nuanced.

These days, I think of mutation rate as analogous to a hyperparameter that affects the learning rate of a machine learning model. Certain settings of the mutation rate knob may allow for faster adaptation than other settings. However, the extent to which mutation rate variation is caused by selection on "how best to set the mutation knob" is poorly understood. While the inference that higher mutation rates in the disease isolates are caused by selection is completely reasonable, the factors that Reviewer 3 raises (indirect selection, i.e. mutation rate hitchhiking with the beneficial mutations that they generate) could be dominating factors here.

As a counterpoint to the results here, see the results reported in my recent paper on divergent mutation rates and biases in Lenski's LTEE (https://academic.oup.com/gbe/article/12/9/1591/5898197). There, completely different mutation rates and biases evolve across replicate populations adapting in parallel to identical abiotic conditions for 30+ years. Based on that, I'm not convinced that selection on "how best to set the mutation knob" is nearly as important as direct selection on traits relevant to reproduction and survival, and concomitant effects on mutation rates and biases. (Although perhaps mutation rates are highly relevant to reproduction and survival in this system!) Hence, I think it will be critical to see whether the pattern reported here will hold, as more pairs of isolates are sampled.

Regardless, I think these data are a valuable contribution to the field, and I find the interpretation of the results well measured after revision.

Reviewer #2: I am satisfied with the authors’ revision and do not have further questions.

Reviewer #3: Thanks for the thoughtful responses and edits to prior reviews. It's a careful and well-executed study that does its best to overcome somewhat limited statistical power. I hope it will motivate other similar tests of this hypothesis.

**Have all data underlying the figures and results presented in the manuscript been provided?**

Reviewer #1: Yes

Reviewer #2: Yes

Reviewer #3: Yes

PLOS authors have the option to publish the peer review history of their article (what does this mean?). If published, this will include your full peer review and any attached files.

Reviewer #1: **Yes: **Rohan Maddamsetti

Reviewer #2: No

Reviewer #3: No

**Data Deposition**

http://datadryad.org/submit?journalID=pgenetics&manu=PGENETICS-D-21-00863R1

**Press Queries**

---

## [Editor Report · Acceptance letter]

2 Nov 2021

PGENETICS-D-21-00863R1 

Mutation rate dynamics reflect ecological change in an emerging zoonotic pathogen 

Dear Dr Murray, 

We are pleased to inform you that your manuscript entitled "Mutation rate dynamics reflect ecological change in an emerging zoonotic pathogen" has been formally accepted for publication in PLOS Genetics! Your manuscript is now with our production department and you will be notified of the publication date in due course.

With kind regards,

Andrea Szabo

PLOS Genetics

On behalf of:
